



**Shrub type dominates the vertical distribution of leaf C:N:P stoichiometry across**
**an extensive altitudinal gradient**
**Wenqiang Zhao[1], Peter B. Reich[2], Qiannan Yu[1,3], Ning Zhao[4], Chunying Yin[1],**
**Chunzhang Zhao[1], Dandan Li[1], Jun Hu[1], Ting Li[1], Huajun Yin[1] and Qing Liu[1]**
[1]CAS Key Laboratory of Mountain Ecological Restoration and Bioresource Utilization
& Ecological Restoration Biodiversity Conservation Key Laboratory of Sichuan
Province, Chengdu Institute of Biology, Chinese Academy of Sciences, Chengdu
610041, China
[2]Department of Forest Resources and Institute on the Environment, University of
Minnesota, Minnesota 55108, USA
[3]Southwest Jiaotong University & Faculty of Geosciences and Environmental
Engineering, Chengdu 611756, China
[4]Cold and Arid Regions Environmental and Engineering Research Institute, Chinese
Academy of Sciences, Lanzhou 730000, China
*Correspondence to:* Qing Liu and Huajun Yin (liuqing@cib.ac.cn; yinhj@cib.ac.cn)



**Abstract.** Understanding the leaf stoichiometric patterns is crucial for improving
predictions on plant responses to environmental changes. Leaf stoichiometry of
terrestrial ecosystems has been widely investigated along latitudinal and longitudinal
gradients. Still, very little is known on the vertical distribution of leaf C:N:P and the
relative effects of environmental parameters, especially for shrubs. Here, we analyzed
the shrub leaf C, N and P patterns in 125 mountainous sites over an extensive altitudinal
gradient (523−4685 m) on the Tibetan Plateau. Results showed that the shrub leaf C
and C:N were 7.3%−47.5% higher than those of other regional and global flora,
whereas the leaf N and N:P were 10.2%−75.8% lower. Leaf C increased with rising
altitude and decreasing temperature, supporting the physiological acclimation
mechanism that high leaf C (e.g., alpine or evergreen shrub) could balance the cell
osmotic pressure and resist freezing. The largest leaf N and high leaf P occurred in
valley region (altitude 1500 m), likely due to the large nutrient leaching from higher
elevations, faster litter decomposition and nutrient resorption ability of deciduous
broadleaf shrub. Leaf N:P ratio further indicated increasing N limitation at higher
altitudes. Interestingly, the drought severity was the only climatic factor positively
correlated with leaf N and P, which was more appropriate for evaluating the impact of
water status than precipitation. Among the shrub ecosystem and functional types (alpine,
subalpine, montane, valley, evergreen, deciduous, broadleaf, and conifer), their leaf
element contents and responses to environments were remarkably different. Shrub type
was the largest contributor to the total variations in leaf stoichiometry, while climate
indirectly affected the leaf C:N:P via its interactive effects on shrub type or soil.
Collectively, the large heterogeneity in shrub type was the most important factor
explaining the overall leaf C:N:P variations, despite the broad climate gradient on the
plateau. Temperature- and drought-induced shift of shrub type distribution will





influence the nutrient accumulation in mountainous shrubs.
**Keywords.** leaf stoichiometry, mountainous shrub, altitudinal gradient, drought
severity, temperature, precipitation, soil nutrient

**1 Introduction**

Ecological stoichiometry examines the interactions among organisms' element
composition and their environments, which provides an effective way to enhance our
understanding of ecosystem function and nutrient cycling (Allen and Gillooly, 2009;
Venterink and Güsewell, 2010). Over the past decades, great attention has been paid to
the leaf stoichiometry of terrestrial plants at regional (Townsend et al., 2007; Matzek
and Vitousek, 2009), national (Han et al., 2011; Sardans et al., 2016), and global scales
(Elser et al., 2000; Reich and Oleksyn, 2004). The leaf macroelements (carbon, nitrogen
and phosphorus) were widely explored to indicate nutrient limitation and its response
to environmental change (Elser et al., 2010). Investigating the interactions among leaf
stoichiometry and the environment along geographic gradients is ctritical to understand
the nutrient cycling process and the development of biogeochemical models.

Nowadays, it is increasingly rare to localize and work on extensive and natural

altitudinal gradient varying from low to high-altitude mountaintops (Nogués-Bravo et
al., 2008). A few studies have investigated the variations of leaf N and P at several
elevations (Soethe et al., 2008; van de Weg et al., 2009; Fisher et al., 2013; Zhao et al.,
2014); however, the scientists reported different altitudinal trends for leaf N and P. For
instance, Soethe et al. (2008) found that the foliar N and P concentrations of trees, herbs
and shrubs were largest at 1900 m than 2400 m and 3000 m in an Ecuadorian montane
forest. van de Weg et al. (2009) observed that the foliar P along an altitudinal transect
(220, 1000, 1500, 1855, 2350, 2990 and 3600 m) from lowland to montane cloud forest

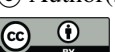



showed no altitudinal trend. Fisher et al. (2013) reported that leaf N and P firstly
increased and then declined with increasing altitudes (200, 1000, 1500 and 3000 m) in
the Peruvian Andes. Zhao et al. (2014) observed that the leaf N and P of 175 plant
species decreased with elevation on the Changbai Mountain (540, 753, 1286, 1812,
2008 and 2357 m). This discrepancy of previous results indicates that the leaf N and P
can vary along different altitudinal ranges at regional scale. Moreover, much less
information was available on the vertical distribution of leaf C. Hence, the more general
patterns of leaf C:N:P along an extensive altitudinal gradient need to be further
understood.
While comprehensive investigations of plant stoichiometry in forestland, grassland,
wetland, and macrophyte ecosystems have emerged (Güsewell and Koerselman, 2002;
He et al., 2006; Townsend et al., 2007; Sardans et al., 2012; Xia et al., 2014), much
fewer studies focused on mountainous shrubs. In China, shrubland is a widely
distributed biome type, covering ~20% of the country. However, information on the
element concentrations of shrubs is very scarce (Piao et al., 2009). Thus, intensive
investigation of shrub stoichiometry can provide detailed information for the growing
global database of plant stoichiometry. As the earth's highest plateau, the Tibetan
Plateau exhibits one of the very few extensive elevational vegetation gradients
remaining in the world (Chen et al., 2013b) (Fig. 1 and Appendix B: Fig. B1). Large
precipitation and temperature gradients along the steep mountains bordering the plateau
to the east lead to a heterogeneous environment. This plateau is also considered as the
China's hotspot ecoregions of biodiversity (Tang et al., 2006). The shrubs here are also
very diverse and widely distributed along altitudinal gradients, which can endure
extreme cold or drought environments. Some shrub species that are unique to this
region have emerged in cold plateau areas (e.g., *Rhododendron telmateium* and





*Quercus monimotricha*). Consequently, the Tibetan Plateau can provide a more general
representation of the stoichiometry of various shrub types, which is an ideal site for
examining the altitudinal patterns and environmental variables influencing shrub
stoichiometry.
The objectives of this study were to (I) analyze the leaf C:N:P stoichiometric
patterns of various shrub types, and (II) clarify the significant factors affecting shrub
stoichiometry across an extensive altitudinal gradient. In this work, we measured the
leaf C, N and P concentrations of 48 shrub species on the Tibetan Plateau. The
geographic, climatic and soil data of sampling sites were recorded. Given that the
Tibetan Plateau encompasses a singular region of high spatial heterogeneity and
complex climatic conditions (Chen et al., 2013b) that may greatly affect shrub nutrient
accumulation, we hypothesized that (I) the overall leaf C:N:P variations would be
dominated by climate, and (II) the shrub leaf element contents would be different from
other terrestrial ecosystems. In addition, plant types and species can greatly affect the
leaf element concentrations (McGroddy et al., 2004). To reveal this effect, all shrubs
were classified into four ecosystem types that located in different vertical vegetation
belts (alpine, subalpine, montane and valley shrub), or three functional types based on
different leaf traits (evergreen broadleaf, evergreen conifer, and deciduous broadleaf
shrubs). Four dominant shrub species (*Rhododendron telmateium* – alpine, *Quercus*
*monimotricha* – subalpine, *Coriaria sinica* – montane, and *Bauhinia brachycarpa* –
valley) were also chosen to assess the leaf patterns at species level.

**2 Materials and Methods**
**2.1 Description of the study area**
Shrub is defined as a small or medium-sized woody plant, which is distinguished from



a tree by its multiple stems and shorter height (below 5 m). Since shrub ecosystems are
mainly distributed in the southeastern margin of the plateau (Appendix B: Fig. B1), we
chose 108 mountainous sites of this region to examine the leaf stoichiometry of shrubs
that included alpine, subalpine and valley areas. Additionally, the 17 neighbouring
mountainous sites on the east of the Tibetan Plateau were selected to provide a
representation of low-altitude montane region. In these areas, shrub is one of the most
important growth forms. Evergreen broadleaf and deciduous broadleaf shrubs are the
primary functional types. The mean annual temperature (MAT) and mean annual
precipitation (MAP) vary from −4.67 to 22.16 ℃, and from 366.3 to 1696.3 mm,
respectively.

Fig. 1 shows the distribution of shrub ecosystem types and sample sites. These sites

contain extensive vertical zonation of shrubs, including alpine (3091−4685 m),
subalpine (2000−4078 m), montane (523−3342 m) and valley shrubs (600−2350 m)
(Appendix A: Table A1). The elevations of four ecosystem types are overlapping
because of the high spatial heterogeneity and diverse vegetation that adapted to
environments at different altitudes on the plateau. Alpine shrub is the main ecosystem
type located above the tree line, while subalpine shrub is distributed in the subalpine
coniferous forest zone (Appendix B: Figs. B2 and B3) (Worboys and Good, 2011).
Montane shrub exists in the evergreen and deciduous broadleaf forests, and valley shrub
occurs in the valley region.

**2.2 Field sampling**
During the growing seasons (from July to August) of 2011−2013, sample collection
was performed in 125 mountainous sites, with shrub coverage more than 30%. At each
site, three plots (5 m × 5 m) were randomly set up, and the distances among different





plots were 5−50 m. For each plot, mature leaves from dominant shrub species of 5−10
individuals were collected and mixed. After litter was removed from the soil surface,
nine 3-cm-diameter soil cores (0−10 cm layer) in each plot were collected and
combined to form one composite sample to account for any heterogeneity resulted from
position. After collection, the leaf samples were oven-dried at 65 ºC, and ground to fine
powders using a ball mill for element analysis. The fresh soils were air-dried, with
visible roots, stones and organic residues removed. Soil samples were sieved through
2-mm meshes before analysis.

**2.3 Geographic and climatic parameters**
The geographic locations (altitude, latitude and longitude) of sample sites were
recorded using a global positioning system. MAT and MAP values were obtained from
the China Meteorological Forcing Dataset (Yang et al., 2010; Chen et al., 2011). The
temporal and spatial resolutions of this dataset were every 3 h and 0.1º × 0.1º in
longitude and latitude from 1981 to 2008.

Considering the mountainous areas exhibit various drought conditions (especially

in valley region), we herein first investigate how leaf stoichiometry varies with drought
index (Reconnaissance Drought Index, RDI). RDI has been widely used in meteorology
to powerfully assess drought severity in arid and semiarid regions (Tsakiris and
Vangelis, 2005). Compared to the other indices (e.g., the Palmer Drought Severity Index
and the Standardized Precipitation Index), the advantages of RDI are its low data
requirements, high resilience and sensitivity to drought events (Khalili et al., 2011).
The standardized form of RDI ($RDI_{st}$) can be calculated via the computation of potential
evapotranspiration (PET) based on the Thornthwaite method (Thornthwaite, 1948). The
detailed calculation process of $RDI_{st}$ for a hydrological year (12-month reference period)

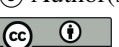



was shown in the Appendix C. In this work, RDI is used to represent $RDI_{st}$. Positive
RDI represent wet period of sample site, whereas negative values indicate dry period.
Using the RDI values, drought severity can be categorized as extreme ($< -2.0$), severe
($-2.0$ to $-1.5$), moderate ($-1.5$ to $-1.0$) or mild ($-1.0$ to $-0.5$) (Vangelis et al. 2013).

**2.4 Element measurements**
The total C and N concentrations of leaf and soil samples were measured by dry
combustion using a Perkin-Elmer 2400 II elemental analyzer (Perkin-Elmer, Inc., USA).
Total P concentrations were determined by the ammonium molybdate method using a
continuous-flow analyser (AutoAnalyzer3 Continuous-Flow Analyzer; Bran Luebbe,
Germany) after $H_2SO_4$-$HClO_4$ digestion for leaves and $H_2SO_4$-$H_2O_2$-HF digestion for
soil (Kuo, 1996). The element concentrations are presented in units of mg g$^{-1}$ dry
weight, and the element ratios are presented on a mass basis. Unfortunately, soil
available nutrient data were not determined, and we were unable to assess their
correlations with leaf elements.

**2.5 Data analysis**
The data were analyzed at two levels: (1) using all the shrub samples together ($n = 125$),
and (2) dividing the shrub dataset into four ecosystem types and four dominant shrub
species that located in different altitudinal belts, or three functional types classified by
different leaf traits. The leaf mineral concentrations and ratios were averaged at the site-
species level to control for pseudoreplication (Han et al., 2011). Because the leaf
element concentrations and ratios were highly skewed (Appendix B: Fig. B4), we
calculated their geometric means, standard deviations and coefficients of variation (CV)
to compare the differences of leaf C:N:P among different shrubs. Besides, the arithmetic



means of shrub leaf stoichiometry were also calculated to compare with prior studies
that only showed arithmetic means.
After all the leaf C:N:P values were $\log_{10}$-transformed to improve the data normality,
there was still no evidence for the test of homogeneity of variances. Therefore, we used
Kruskal-Wallis test (nonparametric) followed by pairwise comparisons to examine the
differences in leaf C:N:P among shrub types.
Partial correlation analysis allows one to distinguish the degree of the direct
correlation between geography (e.g., altitude) and leaf element, with the effect of other
controlling random variables removed (e.g., longitude and latitude). Linear and
nonlinear regressions were utilized to show the variation trends of leaf C:N:P along
climatic and soil gradients. Stepwise multiple regression (SMR) was applied to select
the most influential environmental factors (MAT, MAP, RDI, soil C, soil N and soil P),
and estimate their contributions to leaf stoichiometry.
To evaluate the relative effects of shrub type (ecosystem and functional types), soil
(soil C, N and P) and climate (MAT, MAP, RDI), partial general linear model (GLM)
analysis was applied. Partial GLM separates the total variance explained by different
factors into the independent effect of each factor and their interactive effects (Heikkinen
et al., 2005). The statistical analyses were conducted with SPSS v20 (SPSS Inc., USA),
Origin 8.0 (OriginLab Co., USA) and R 2.15.2.

**3 Results**
**3.1 Variations of leaf C, N, P and C:N:P ratios**
Leaf C, N and P contents for all shrubs ranged from 358.2 to 539.8, 9.7 to 39.4 and 0.69
to 3.43 mg g$^{-1}$, respectively (Supplementary material). The element ratios varied
greatly with a range of 11.7−46.9 for C:N, 113.8−646.5 for C:P, and 2.86−22.16 for N:P.



The geometric means were 468.9 mg g$^{-1}$ for C, 18.6 mg g$^{-1}$ for N and 1.50 mg g$^{-1}$ for
P, while those for C:N, C:P and N:P were 25.3, 312.0 and 12.3, respectively.

The geometric means of leaf C for alpine and subalpine shrubs were 481.7 and

477.6 mg g$^{-1}$, respectively, which were higher than those of montane and valley shrubs
($P < 0.001$, Table 1 and Appendix A: Table A2). Leaf N of valley shrub was the highest
among the ecosystem types, while the leaf P contents of subalpine and montane shrubs
were lower than alpine shrub ($P < 0.001$). Leaf elements also varied markedly across
different functional types. Deciduous broadleaf shrub showed the lowest leaf C,
whereas its leaf N was the largest ($P < 0.01$). Leaf P did not show significant difference
among the three functional types ($P = 0.323$). For dominant shrub species, the leaf
C:N:P in *Rhododendron telmateium* (alpine), *Quercus monimotricha* (subalpine),
*Coriaria sinica* (montane), and *Bauhinia brachycarpa* (valley) followed similar trends
to those in corresponding shrub ecosystem types.

The relative variability of leaf nutrients can be demonstrated by CV. Leaf P of all

samples had the greatest variation (37.0%), followed by N (30.4%) and C (6.3%). The
relative variability of leaf elements for each shrub type also showed the similar trends.

**3.2 Altitudinal patterns of leaf stoichiometry**
Using partial correlation analysis (Appendix A: Table A3), we found that leaf C
increased with the increase of altitude ($P < 0.001$), while the leaf N and P did not show
clear altitudinal trend ($P = 0.287$ and 0.154). The highest leaf N and P were distributed
at altitude of about 1500 m which belonged to valley shrub (Fig. 2).

The relationships between the leaf stoichiometry of shrub types and elevations

exhibited different patterns. For example, altitude was not correlated with the leaf C of
alpine and subalpine shrubs. On the contrary, leaf N and P showed obvious altitudinal




trends for subalpine and deciduous broadleaf shrubs ($P < 0.01$).

**3.3 Climatic influence on leaf stoichiometry**
MAP was not linearly or nonlinearly correlated with most of the leaf C:N:P traits
(Appendix B: Fig. B5). MAT was negatively correlated with leaf C, while the leaf P,
C:P and N:P were quadratically correlated with MAT ($P < 0.001$). As RDI increased
(i.e., wetter conditions), leaf N and P became significantly larger ($P < 0.001$). When the
three climatic factors were analyzed by SMR, MAP was excluded from all the analysis
(Table 2). MAT was negatively related with leaf C ($P < 0.001$), and only RDI was
entered into the SMR equations for leaf N and P.

The climatic factors showed large heterogeneity across different shrub types and

species (Appendix A: Tables A4−A6). For instance, the leaf N or P of alpine, valley
shrubs and *Quercus monimotricha* were correlated with MAT or MAP ($P < 0.05$), while
those of montane, evergreen conifer shrubs, *Rhododendron telmateium* and *Coriaria*
*sinica* were not affected by climate ($P > 0.05$). It indicated that the specific shrub type
or species exhibited diverse leaf C:N:P trends along climatic gradient and change
greatly to adapt to different habitats.

**3.4 Soil influence on leaf stoichiometry**
Plants take up most of the nutrients directly from soils. As usually observed elsewhere,
our results revealed positive correlations between the leaf and soil stoichiometry for C,
P, C:P and N:P ($P < 0.05$) (Appendix B: Fig. B6, Table 2). The leaf N and C:N of all
shrubs were not correlated with those of 0−10 cm soil layer, probably owing to
confounding effects of other variables along geographic gradients. Interestingly, the
leaf N, P, C:N, C:P and N:P of evergreen broadleaf shrub were only correlated with soil



variables (Appendix A: Table A5), indicating the leaf nutrients in evergreen broadleaf
shrub were mainly affected by root uptake from soils. By contrast, the soil elements
were not limiting factors for the leaf element levels in montane shrub and *Coriaria*
*sinica* (Appendix A: Tables A4 and A6).

**3.5 Relative effects of shrub type, soil and climate**
The three factors (shrub type, soil and climate) together accounted for 45.2%−54.5%
of the six leaf C:N:P traits (Fig. 3). The total effect of shrub type ($t+ct+st+cst$) showed
the largest contribution to the variations in leaf stoichiometry (37.9%−53.9%). The
independent effect of shrub type ($t$, 19.2%−44.7%) was also greater than those of soil
($s$, 0%−34.7%) and climate ($c$, 0%−0.4%). Soil exhibited the largest independent
contribution ($s$, 34.7%) to the variation of leaf P. Climate ($c+ct+cs+cst$, 0%−16.1%)
contributed to leaf stoichiometry mainly via the interactive effects between climate and
shrub type ($ct$) or among the three factors ($cst$). The negative value (e.g., $cs = -27.1\%$
for leaf P) indicated suppressive interactive effects of climate and soil.

**4 Discussion**
A few studies have found that the leaf elements of plants varied largely with altitudes
in different mountainous regions (Soethe et al., 2008; van de Weg et al., 2009; Fisher
et al., 2013). However, the relative effects of shrub types and environmental variables
on leaf elements have not yet been addressed. The statistical analysis proved that shrub
type explained the largest fraction of the leaf C:N:P variations, and the leaf element
levels differed from other terrestrial ecosystems. This work provides important
information on the specific leaf patterns of various shrub types and species over a large
altitudinal gradient.




## 4.1 Different leaf C:N:P levels of shrubs on the plateau

The leaf C, N and P of shrubs on the Tibetan Plateau confirmed our hypothesis (II) that
they were different from those at regional, national and global levels (Table 3),
suggesting mountainous shrubs had different ways in allocating nutrients. The
arithmetic means of shrub leaf C and C:N were 7.3%−47.5% greater than those of other
regional and global flora ($P < 0.05$), whereas the mean leaf N and N:P were 10.2%−75.8%
lower ($P < 0.05$, except herbaceous species in central England) (Thompson et al., 1997;
Elser et al., 2000; Campo and Dirzo, 2003; Reich and Oleksyn, 2004; Han et al., 2005;
Tibbets and Molles, 2005; He et al., 2006; Townsend et al., 2007; Zheng and Shangguan,
2007; Chen et al., 2013a). The arithmetic mean of shrub leaf P (1.60 mg g$^{-1}$) were within
the range of those reported in other regions (0.82−2.70 mg g$^{-1}$). In this study, the
altitudes (523−4685 m) were much wider and higher than those investigated in other
terrestrial ecosystems (Soethe et al., 2008; van de Weg et al., 2009; Fisher et al., 2013).
Two classical hypotheses may account for this phenomenon. On the one hand, based
on the plant physiological acclimation mechanism, it is likely that more non-structural
C (e.g., starch, low molecular weight sugars and storage lipid) may accumulate in leaf
(e.g., alpine shrub) to balance the osmotic pressure of cells and resist freezing (Hoch et
al., 2002; Hoch and Körner, 2012). On the other hand, according to the Biogeochemical
Hypothesis, low temperatures in these areas could limit soil microbe activity (Reich
and Oleksyn, 2004). It may lead to slower decomposition of soil organic matter, and
probably depress available N uptake by roots.
Among various mineral elements, N and P are considered the major growth-
constraining nutrients (Koerselman and Meuleman, 1996). The shrub growth was
relatively limited by N (mean leaf N:P = 12.8). The decreased leaf N:P with lower



temperatures (Table 2) further suggested that the growth of shrubs at higher altitudes
are more limited by N. However, Han et al. (2005) reported that the 547 plant species
in China were strongly constrained by P, with mean leaf N:P (16.3) significantly higher
than those in global flora and shrubs in this work ($P < 0.05$, Table 3). It indicated that
the assessment of nutrient limitation at large scale could not reflect the pattern in
mountainous areas.

The CV patterns among leaf elements are consistent with the Stability of Limiting

Elements Hypothesis (Sterner and Elser, 2002). It is known that plant nutrient (e.g., C)
that required at a high concentration should show a small variation and lower sensitivity
to the environment. Leaf C was less variable than leaf N and P, suggesting leaf C had
stronger stoichiometric homeostasis. The CV value of shrub leaf C (6.3%) was smaller
than those of trees, herbs and shrubs (6.9%−28.0%) in other regions, whereas those of
shrub leaf N (30.4%) and P (37.0%) were within the range of other ecosystems (N:
11.0%−50.5%; P: 13.0%−44.0%) (Tibbets and Molles, 2005; He et al., 2006; Zheng
and Shangguan, 2007; Ladanai et al., 2010). Consequently, the high C accumulation
capacity of shrub is less sensitive to the complex climate conditions on the plateau.

**4.2 Relative influences of the environment and shrub type**
Precipitation, temperature and soil can affect leaf elements via changing element
allocation among plant organs, altering plant metabolism or influencing nutrient uptake
by roots (Ordoñez et al., 2009). In addition to MAP, MAT and soil nutrient, we first
added RDI to examine the effect of local drought extent. Among the four environmental
parameters, it is interesting to note that RDI was positively correlated with leaf N and
P (Table 2). By contrast, MAP was not correlated with all the leaf C:N:P traits. This was
inconsistent with previous reports that MAP played an important role for the leaf





elements of different vegetation types (Santiago et al., 2004; Han et al., 2011). Firstly,
among the 125 sampling sites, only 7 sites belonged to the severe and extreme drought
regions (RDI < –1.5). The water conditions of other sites were mild, slight drought, or
wet (–1.0 < RDI < 2), which may be suitable for shrub growth and could not become a
limiting factor. Secondly, it is proposed that MAP could not accurately reflect the real
water situation due to different temperatures and evapotranspiration rates. RDI may be
more appropriate for evaluating the impact of water status. The wetter climatic
conditions (i.e., larger RDI) could provide more soluble N and P in soil and enhance
the nutrient transportation of shrub.

360   Increased MAT was found to be related with the decrease of leaf C in the SMR

analysis. This result was in agreement with a meta-analysis of C stores conducted in 13
different global mountains (Hoch and Körner, 2012). The large MAT gradient (–4.67
ºC to 22.16 ºC) on the plateau could strongly affect the shrub photosynthesis process.
Shrub species at higher elevations probably need to protect themselves against low
temperatures and make osmotic adjustments via increasing leaf C contents (Millard et
al., 2007). By contrast, MAT could not account for the leaf N in shrubs, which was
inconsistent with the opinion that leaf N contents are usually affected by temperature
(Wright et al., 2005). This unexpected phenomenon may result from the large
heterogeneity in N uptake capacities of different shrub species along the climatic
gradients (CV of leaf N reaches up to 30.4%). Moreover, the drought severity was so
dominant in leaf N that it may override any possible underlying temperature effect.

372   Soil nutrient was the most significant environmental factor for leaf P, C:P and N:P.

Appendix B: Fig. B6 also exhibits the closest relationship between soil P and leaf P ($P$
< 0.001). It was known that P mainly originates from the soil via rock weathering
(Walbridge et al., 1991). Moreover, all the soil C:P ratios were less than 200 (implies



376 net mineralization in soil), confirming the soil may provide sufficient soluble P (Bui

377 and Henderson, 2013).

378  Climate, soil nutrient and vegetation type can together influence plant mineral

379 biogeography in complex ways, while significant collinearities among these factors

380 may potentially obscure their true impacts (Han et al., 2011). Here we used partial GLM

381 regressions to separate the total variance into the independent effect of each factor and

382 their interactive effects (Heikkinen et al., 2005). We found that the independent effect

383 ($t$) of shrub type was the largest contributor to explain the leaf element variations (Fig.

384 3). For climate, however, the result disagreed with our hypothesis (I) that the leaf C:N:P

385 variations would be dominated by climate. The independent effect of climate ($c$) was

386 small, and climate mainly affected the leaf C:N:P via its interactive effects on shrub

387 type ($ct$) or among the three factors ($cst$). Combined with SMR analysis (Table 2), this

388 finding suggests that climate-induced (e.g., temperature and drought) changes of shrub

389 distribution may affect the leaf nutrient contents. Soil nutrient ($s$) accounted for large

390 parts of the variations in leaf P and C:P ratio, which was ascribed to the coupled

391 relationships between soil P and plant P (Walbridge et al., 1991).

393 **4.3 Large heterogeneity in leaf C:N:P patterns among various shrubs**

394 To our knowledge, the leaf C:N:P patterns among different shrub types and species have

395 not been sufficiently evaluated before. Our analysis suggests the leaf element contents

396 and their responds to environments were highly heterogeneous among shrub types and

397 species, providing further evidence that the large heterogeneity in shrub nutrient uptake

398 capacities and physiological adaptation to environments governed the leaf nutrient

399 variations.

400  For instance, alpine and subalpine shrubs had higher leaf C than the other two





ecosystem types (Table 1 and Appendix A: Table A2), and this trend was consistent
with the cold acclimation mechanism as discussed previously (Hoch and Körner, 2012).
Valley shrub possessed the greatest leaf N and high leaf P, especially at altitude of about
1500 m. This result could be explained as follows: (I) larger nutrient deposition in the
valleys may result from accumulated sediment, nutrient leaching, or runoff from higher
elevations (average MAP reaches up to 882.1 mm); (II) the functional type in valley
was mostly short-lived, fast-growing deciduous broadleaf shrub (e.g., *Bauhinia*
*brachycarpa*, Table 1), which exhibited faster litter decomposition and nutrient
resorption abilities than long-lived, slow-growing evergreen types (Güsewell and
Koerselman, 2002; Diehl et al., 2003); and (III) the MAT values of valley sites were
relatively higher than montane, subalpine and alpine sites ($P < 0.05$, Appendix A: Table
A1), indicating faster organic matter decomposition as predicted by the Biogeochemical
Hypothesis (Aerts and Chapin, 1999). It should be also noted that the alpine shrub
exhibited higher leaf N and P than subalpine and montane shrubs located in low-altitude
regions ($P < 0.001$). This result agreed with the Temperature-Plant Physiological
Hypothesis (Weih and Karlsson, 2001; Zhang et al., 2017). In high-altitude area, the
growing season was short, and accompanied by lower temperature. Hence, shrubs
might increase their nutrient absorption to compensate for lower enzyme efficiency and
metabolic rate.
Large differences in leaf elements also occurred across functional types (Table 1
and Appendix A: Table A2). Leaf C contents in evergreen broadleaf and evergreen
conifer shrubs were higher than deciduous broadleaf shrub, agreeing with the higher
non-structural C accumulated in evergreen shrub leaves (average altitude: 3430 m) to
resist freezing than deciduous shrub (average altitude: 2343 m). On the contrary, leaf N
was larger in deciduous broadleaf shrub than in evergreen shrub types ($P < 0.01$). This



result was ascribed to higher nutrient resorption in deciduous species than in evergreen
species (Güsewell and Koerselman, 2002). Moreover, lower leaf C:N and C:P ratios
were observed in deciduous shrub than evergreen shrubs ($P < 0.05$), further indicating
faster litter decomposition process of deciduous shrub (Bui and Henderson, 2013).
These differences of element levels among functional types indicate the variations in
leaf nutrient acquisition abilities.
The leaf traits of shrub ecosystem, functional types and species with respect to four
environmental variables differed from each other (Appendix A: Tables A4−A6).
Specifically, the leaf N and P contents of montane, evergreen conifer shrubs and
*Rhododendron telmateium* were not correlated with climate or soil, whereas those of
evergreen broadleaf, deciduous broadleaf and *Bauhinia brachycarpa* exhibited positive
relationships with soil or RDI. It revealed that the diverse shrubs showed great
heterogeneity in their responses to water status and soil nutrients. Interestingly, the leaf
N of evergreen broadleaf shrub had the closest correlation with soil N ($P < 0.01$).
Meanwhile, the evergreen broadleaf shrub was largely limited by N (mean leaf N:P =
11.5) (Table 1), suggesting the growth of this N-limiting shrub may be highly sensitive
to soil N contents.

**5 Conclusions**
This work was the first field investigation of the leaf C:N:P stoichiometry of different
shrub types along an extensive altitudinal range, providing important data for future
research on global C, N and P cycling. Results highlight that different shrub leaf C:N:P
contents and ratios emerged compared to other terrestrial ecosystems, and the leaf
C:N:P variations were primarily explained by shrub type. This phenomenon is likely
due to the large heterogeneity in nutrient uptake and physiological adaptation to



extreme environments across various shrubs on the plateau. However, the underlying
physiological mechanisms of specific shrub type or species require further examination.
Our findings also indicated that the drought severity was the key climatic factor
correlated with leaf N and P, which should be integrated into future biogeochemical
models of element cycling. We should pay attention to the N shortage problem to
improve the growth of shrubs. Global changes in vegetation distribution, temperature
and drought severity will strongly affect the spatial patterns of shrub nutrient pools and
ecosystem functioning.

*Data availability.* Raw data are available in the Supplementary material.

*Competing interests.* The authors declare that they have no conflict of interest.

*Acknowledgements.* This work was supported by the National Key R&D Program of
China (2017YFC0505000), the National Natural Science Foundation of China
(31500445, 31400424), the Frontier Science Key Research Programs of the Chinese
Academy of Sciences (QYZDB-SSW-SMC023), and the CAS "Light of West China"
Program (Y6C2051100). The authors would like to thank Prof. Shilong Piao and Ph.D.
student Hui Yang from Peking University for calculating MAP and MAT data. The
China Meteorological Forcing Dataset used in this study was developed by Data
Assimilation and Modeling Center for Tibetan Multi-spheres, Institute of Tibetan
Plateau Research.

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

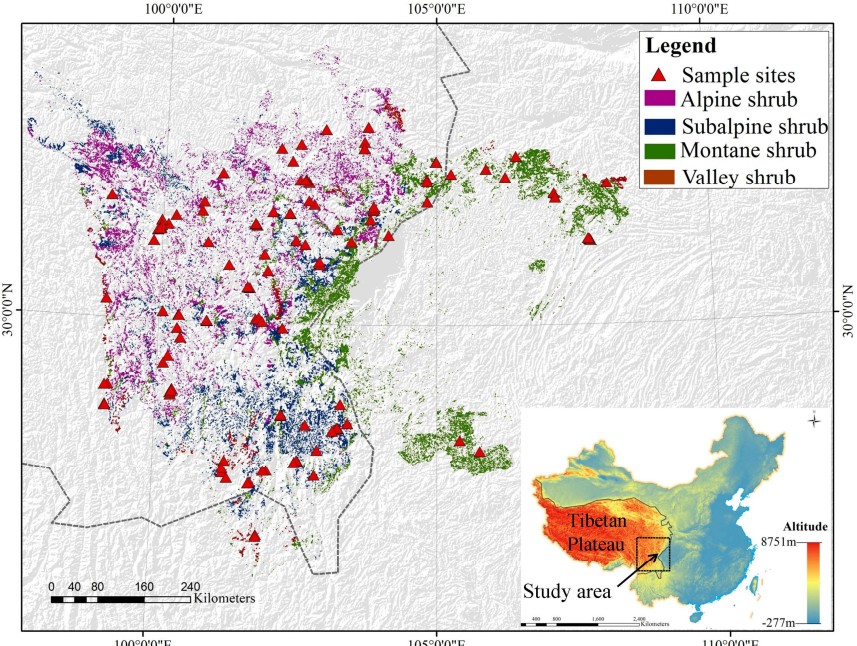


**Figure 1.** Locations of the southeastern Tibetan Plateau and 125 sample sites in

mountainous areas. The purple, blue, green, and brown areas stand for the distribution

of alpine, subalpine, montane and valley shrub types, respectively.
















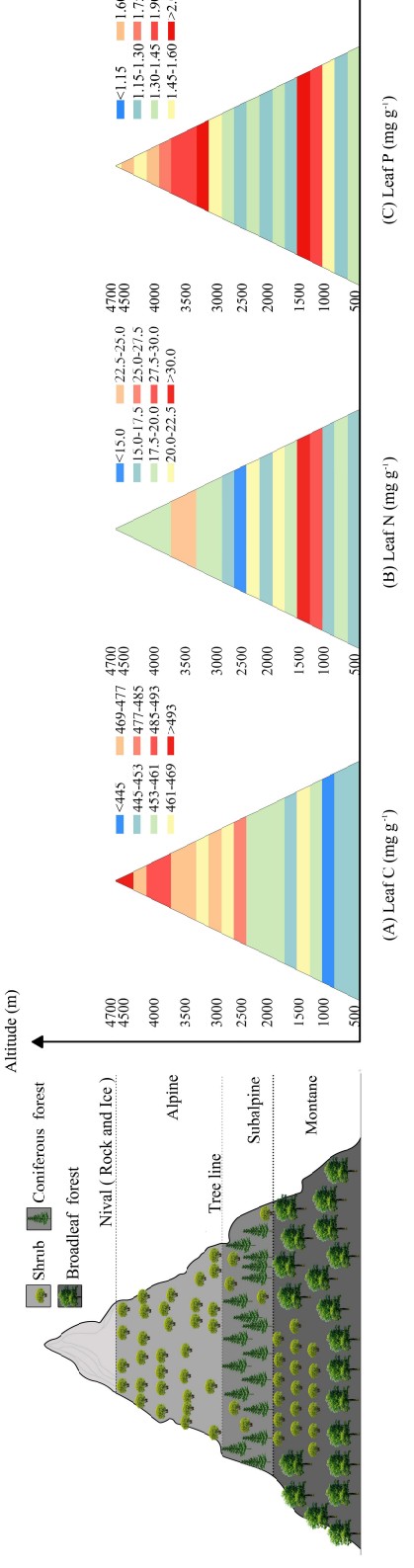

**Figure 2.** Vertical distribution of leaf C, N and P contents of mountainous shrubs on the Tibetan Plateau. Alpine shrub occurs at relatively higher

altitude (3091–4685 m, above tree line), followed by subalpine shrub (2000–4078 m, coniferous forest zone), montane shrub (523–3342 m,

broadleaf forest zone) and valley shrub (600–2350 m, valley region).











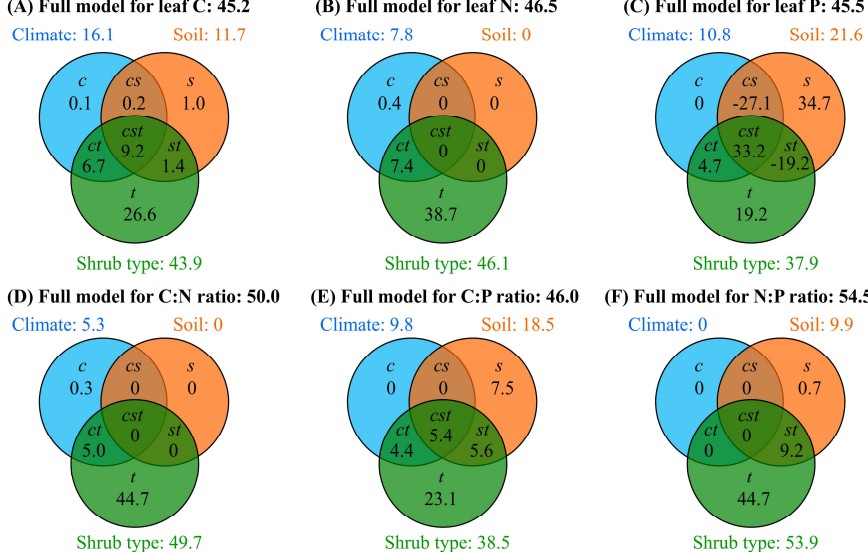

**Figure 3.** Summary of the partial general linear models (GLM) for the effects ($R^2$, %) of climate, soil and shrub type on leaf stoichiometry. *c*, *s*, and *t* represent the independent effects of climate, soil, and shrub type, respectively; *cs*, *ct*, and *st* are the interactive effects between climate and soil, climate and shrub type, soil and shrub type, respectively; *cst* denotes the interactive effect among the three factors. Leaf element concentrations and ratios were log$_{10}$-transformed before analysis. Insignificant climatic or soil variables were not included in the partial GLM analysis. Shrub type stands for the combination of ecosystem type and functional type (e.g., montane deciduous broad-leaf shrub).



**Table 1.** Geometric means and standard deviations of leaf element concentrations and ratios for all shrub samples, and specific shrub type and species on the Tibetan Plateau. Ecosystem types include shrubs that located at different altitudinal belts. Functional types are classified based on different leaf traits. Coefficients of variation (CV, %) are in parentheses. $n$ represents the number of shrub samples. Comparisons of geometric means of leaf C:N:P among shrub types and species ($P$ values) were shown in Appendix A: Table A2.

| | | $n$ | Leaf C (mg g$^{-1}$) | Leaf N (mg g$^{-1}$) | Leaf P (mg g$^{-1}$) | Leaf C:N | Leaf C:P | Leaf N:P |
|---|---|---|---|---|---|---|---|---|
| **All samples** | | 125 | 468.9 ± 29.8 (6.3%) | 18.6 ± 5.9 (30.4%) | 1.50 ± 0.59 (37.0%) | 25.3 ± 7.8 (29.7%) | 312.0 ± 113.5 (34.2%) | 12.3 ± 3.5 (27.5%) |
| **Ecosystem type** | Alpine shrub | 58 | 481.7 ± 31.9 (6.6%) | 19.7 ± 5.8 (28.5%) | 1.75 ± 0.56 (31.0%) | 24.5 ± 8.1 (31.6%) | 276.0 ± 83.1 (28.8%) | 11.3 ± 2.5 (21.6%) |
| | Subalpine shrub | 20 | 477.6 ± 12.2 (2.6%) | 14.7 ± 3.9 (25.9%) | 1.24 ± 0.42 (32.6%) | 32.5 ± 6.8 (20.3%) | 386.4 ± 114.2 (28.3%) | 11.9 ± 2.1 (17.7%) |
| | Montane shrub | 30 | 448.3 ± 18.8 (4.2%) | 17.6 ± 3.5 (19.3%) | 1.23 ± 0.49 (37.1%) | 25.4 ± 5.1 (19.8%) | 357.3 ± 117.7 (31.3%) | 14.1 ± 4.0 (27.3%) |
| | Valley shrub | 17 | 452.9 ± 24.9 (5.5%) | 21.8 ± 7.8 (34.0%) | 1.56 ± 0.71 (41.9%) | 20.8 ± 7.6 (34.8%) | 290.1 ± 132.3 (42.0%) | 14.0 ± 4.7 (31.0%) |
| **Functional type** | Evergreen broadleaf | 55 | 485.3 ± 29.9 (6.2%) | 16.5 ± 4.7 (27.5%) | 1.44 ± 0.50 (33.1%) | 29.4 ± 7.2 (23.6%) | 337.6 ± 103.3 (29.2%) | 11.5 ± 2.5 (21.3%) |
| | Evergreen conifer | 6 | 488.3 ± 16.3 (3.3%) | 13.7 ± 3.9 (27.9%) | 1.34 ± 0.35 (25.6%) | 35.6 ± 8.4 (23.0%) | 363.5 ± 77.4 (20.8%) | 10.2 ± 1.5 (14.6%) |
| | Deciduous broadleaf | 64 | 453.5 ± 20.8 (4.6%) | 21.1 ± 5.9 (27.0%) | 1.58 ± 0.67 (39.3%) | 21.5 ± 5.5 (24.7%) | 287.4 ± 121.1 (39.1%) | 13.4 ± 4.0 (28.6%) |
| **Dominant species** | *Rhododendron telmateium* | 19 | 500.4 ± 12.7 (2.5%) | 19.0 ± 4.3 (22.1%) | 1.63 ± 0.37 (22.4%) | 26.3 ± 5.7 (21.2%) | 306.9 ± 68.7 (21.9%) | 11.6 ± 1.6 (13.4%) |
| | *Quercus monimotricha* | 5 | 464.7 ± 8.0 (1.7%) | 14.9 ± 4.0 (26.4%) | 1.33 ± 0.42 (30.9%) | 31.1 ± 8.9 (27.6%) | 350.5 ± 93.5 (25.9%) | 11.3 ± 1.2 (10.5%) |
| | *Coriaria sinica* | 6 | 426.3 ± 10.2 (2.4%) | 18.4 ± 2.9 (15.4%) | 1.09 ± 0.21 (19.2%) | 23.2 ± 3.2 (13.5%) | 391.5 ± 74.1 (18.7%) | 16.9 ± 2.2 (13.1%) |
| | *Bauhinia brachycarpa* | 3 | 443.5 ± 1.2 (0.3%) | 24.7 ± 2.4 (9.6%) | 1.45 ± 0.28 (19.5%) | 18.0 ± 1.9 (10.3%) | 306.9 ± 63.0 (20.2%) | 17.1 ± 1.8 (10.8%) |





**Table 2.** Model summary for the stepwise multiple regression (SMR) of leaf element

concentrations and ratios of all shrub samples on climatic and soil variables (MAT, MAP,

RDI, soil element and ratio).

| Leaf element | Adj. $R^2$ Full mode | Partial regression coefficient | | | | Contribution of predictor (%) | | | |
|---|---|---|---|---|---|---|---|---|---|
| | | MAT | MAP | RDI | Soil | MAT | MAP | RDI | Soil |
| C | 0.181 | −0.001*** | − | − | 0.001* | 61.6 | − | − | 38.4 |
| N | 0.097 | − | − | 0.037*** | − | − | − | 100 | − |
| P | 0.214 | − | − | 0.034** | 0.138*** | − | − | 40.8 | 59.2 |
| C:N | 0.085 | − | − | −0.036** | − | − | − | 100 | − |
| C:P | 0.141 | − | − | −0.043*** | 0.001** | − | − | 54.8 | 45.2 |
| N:P | 0.060 | 0.004* | − | − | 0.015* | 47.8 | − | − | 52.2 |

*, **, and *** denote significance at the 0.05, 0.01, 0.001 test level, respectively. For

partial regression coefficients, "+" indicates a positive correlation and "−" indicates a

negative correlation. MAT, mean annual temperature; MAP, mean annual precipitation;

RDI, standardized form of Reconnaissance Drought Index. Soil represents

corresponding soil element or ratio relative to leaf element or ratio. Leaf element

concentrations and ratios were $log_{10}$-transformed before analysis.



**Table 3.** Comparison of arithmetic means of leaf C, N, P and C:N:P ratios between the shrubs on the Tibetan Plateau and other regional or global
plants using Kruskal-Wallis test followed by pairwise comparisons. Standard deviations are in parentheses.

| Study area | Leaf C (mg g⁻¹) | Leaf N (mg g⁻¹) | Leaf P (mg g⁻¹) | Leaf C:N | Leaf C:P | Leaf N:P | Reference |
|---|---|---|---|---|---|---|---|
| Shrubs on the Tibetan Plateau, southwestern China | 469.8 (29.8) | 19.4 (5.9) | 1.60 (0.59) | 26.4 (7.8) | 331.7 (113.5) | 12.8 (3.5) | This study |
| Shrubs in the Loess Plateau, central and northern China | 437.0 (36.0)* | 24.7 (8.2)* | 1.55 (0.44) | 19.7 (6.8)* | 302.0 (84.0)* | 16.1 (4.1)* | Zheng and Shangguan, 2007 |
| Woody plants, eastern China | – | 23.2 (7.2)* | 1.59 (0.84) | – | – | 17.6 (7.2)* | Chen et al., 2013a |
| Grassland biomes, China | 438.0 (30.2)* | 27.6 (8.6)* | – | 17.9 (5.7)* | – | – | He et al., 2006 |
| Chinese flora | – | 20.2 (8.4) | 1.46 (0.99) | – | – | 16.3 (9.3)* | Han et al., 2005 |
| Tropical dry forests, Mexico | – | 21.3 (4.5) | 1.15 (0.46) | – | – | 22.2 (11.4)* | Campo and Dirzo, 2003 |
| Tropical rain forests, Brazil, Costa Rica | – | 21.6 (5.6)* | 0.82 (0.34)* | – | – | 28.6 (8.6)* | Townsend et al., 2007 |
| Herbaceous species, central England | – | 27.8 (9.9)* | 2.70 (1.52)* | – | – | 10.7 (2.8)* | Thompson et al., 1997 |
| Dominant riparian trees along the Middle Rio Grande, USA | 463.0 (0.8) | 31.0 (8.0)* | 1.50 (4.10) | 18.4 (4.2)* | 1010 (560)* | 53.0 (21.0)* | Tibbets and Molles, 2005 |
| Global flora | – | 20.1 (8.7) | 1.77 (1.12) | – | – | 13.8 (9.5) | Reich and Oleksyn, 2004 |
| Global flora | 464.0 (32.1) | 20.6 (12.2) | 1.99 (1.49)* | 22.5 (10.6)* | 232.0 (145.0)* | 12.7 (6.8) | Elser et al., 2000 |

* denote significant difference in leaf element traits between the shrubs on the Tibetan Plateau and other regional or global plants at the 0.05 level.




**Appendix A: Additional tables**
**Table A1.** Geographical, climatic and soil nutrient information of different shrub types and representative dominant shrub species on the Tibetan Plateau.

| Shrub type | Altitude (m) | MAP (mm) | MAT (°C) | RDI | Soil C (mg g⁻¹) | Soil N (mg g⁻¹) | Soil P (mg g⁻¹) |
|---|---|---|---|---|---|---|---|
| **Ecosystem type** | | | | | | | |
| Alpine shrub | 3091~4685 | 366.3~1013.4 | -4.25~13.88 | -1.36~2.02 | 19.0~167.1 | 1.41~11.95 | 0.58~2.05 |
| Subalpine shrub | 2000~4078 | 459.8~1008.5 | -4.67~11.98 | -1.71~-0.08 | 17.2~106.2 | 1.16~8.11 | 0.26~2.01 |
| Montane shrub | 523~3342 | 490.0~1555.4 | 0.11~16.97 | -1.64~1.75 | 3.5~92.2 | 0.65~8.51 | 0.18~1.77 |
| Valley shrub | 600~2350 | 373.5~1696.3 | 2.35~22.16 | -1.64~1.38 | 6.1~89.2 | 0.58~7.69 | 0.14~1.51 |
| **Functional type** | | | | | | | |
| Evergreen broadleaf | 627~4685 | 366.3~1555.4 | -4.67~22.16 | -1.71~1.76 | 8.9~161.8 | 1.00~11.95 | 0.23~2.01 |
| Evergreen conifer | 2145~4378 | 471.8~761.8 | 0.12~10.73 | -1.33~1.54 | 23.0~110.3 | 2.23~8.32 | 0.46~1.64 |
| Deciduous broadleaf | 523~4212 | 373.5~1696.3 | -4.25~21.51 | -1.64~2.02 | 3.5~167.1 | 0.58~11.02 | 0.14~2.05 |
| **Representative species** | | | | | | | |
| *Rhododendron telmateium* | 3624~4685 | 366.3~993.8 | -4.25~13.88 | -1.36~1.54 | 26.7~101.2 | 2.07~8.13 | 0.61~1.60 |
| *Quercus monimotricha* | 2000~3325 | 646.5~1008.5 | 0.27~7.81 | -1.53~-0.86 | 36.1~86.7 | 2.75~5.24 | 0.69~1.55 |
| *Coriaria sinica* | 540~3156 | 720.7~1435.7 | 6.74~15.31 | -1.42~-0.43 | 3.5~46.2 | 0.76~2.17 | 0.39~0.88 |
| *Cotinus coggygria* | 600~2011 | 373.5~1435.7 | 2.59~15.31 | -1.64~-0.43 | 16.9~60.8 | 1.37~4.19 | 0.51~0.73 |

MAP, MAT and RDI indicate mean annual precipitation, mean annual temperature and Reconnaissance Drought Index, respectively.





**Table A2.** Comparisons of geometric means of leaf stoichiometry using Kruskal-Wallis test followed by pairwise comparisons.

| Comparison pair | Leaf C | Leaf N | Leaf P | Leaf C:N | Leaf C:P | Leaf N:P |
|---|---|---|---|---|---|---|
| **Ecosystem type** | | | | | | |
| Alpine vs Subalpine | P = 1.000 | P < 0.001 | P < 0.001 | P < 0.01 | P < 0.01 | P = 1.000 |
| Alpine vs Montane | P < 0.001 | P = 0.668 | P < 0.001 | P = 1.000 | P < 0.01 | P < 0.01 |
| Alpine vs Valley | P < 0.001 | P = 1.000 | P = 1.000 | P = 0.612 | P = 1.000 | P < 0.01 |
| Subalpine vs Montane | P < 0.001 | P = 0.118 | P = 1.000 | P < 0.05 | P = 1.000 | P = 0.226 |
| Subalpine vs Valley | P < 0.05 | P < 0.001 | P = 0.229 | P < 0.001 | P = 0.096 | P = 0.113 |
| Montane vs Valley | P = 1.000 | P = 0.168 | P = 0.210 | P = 0.335 | P = 0.396 | P = 1.000 |
| **Functional type** | | | | | | |
| Evergreen broadleaf vs Evergreen conifer | P = 1.000 | P = 0.434 | P > 0.05 | P = 0.484 | P = 1.000 | P = 0.469 |
| Evergreen broadleaf vs Deciduous broad-leaf | P < 0.001 | P < 0.001 | P > 0.05 | P < 0.001 | P < 0.05 | P < 0.05 |
| Evergreen conifer vs Deciduous broad-leaf | P < 0.01 | P < 0.01 | P > 0.05 | P < 0.001 | P = 0.299 | P < 0.05 |
| **Dominant species** | | | | | | |
| *Rhododendron telmateium* vs *Quercus monimotricha* | P = 0.081 | P > 0.05 | P = 0.435 | P = 1.000 | P > 0.05 | P = 1.000 |
| *Rhododendron telmateium* vs *Coriaria sinica* | P < 0.001 | P > 0.05 | P < 0.01 | P = 1.000 | P > 0.05 | P < 0.01 |
| *Rhododendron telmateium* vs *Bauhinia brachycarpa* | P < 0.05 | P > 0.05 | P = 1.000 | P = 0.061 | P > 0.05 | P < 0.05 |
| *Quercus monimotricha* vs *Coriaria sinica* | P = 0.880 | P > 0.05 | P = 1.000 | P = 0.366 | P > 0.05 | P < 0.05 |
| *Quercus monimotricha* vs *Bauhinia brachycarpa* | P = 1.000 | P > 0.05 | P = 1.000 | P < 0.05 | P > 0.05 | P = 0.076 |
| *Coriaria sinica* vs *Bauhinia brachycarpa* | P = 1.000 | P > 0.05 | P = 0.841 | P = 0.784 | P > 0.05 | P = 1.000 |

Differences were statistically significant at the 0.05 level. Leaf element concentrations and ratios were log$_{10}$-transformed before analysis.



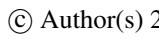

**Table A3.** Partial correlation coefficients between the geographic variables and leaf
stoichiometry of all shrubs and different shrub types.

| | Variable | Leaf C | Leaf N | Leaf P | Leaf C:N | Leaf C:P | Leaf N:P |
|---|---|---|---|---|---|---|---|
| **All shrubs** | Altitude | 0.328*** | −0.097 | 0.129 | 0.163 | −0.066 | −0.252** |
| | Longitude | 0.028 | −0.173 | −0.088 | 0.174 | 0.089 | −0.069 |
| | Latitude | −0.088 | 0.350*** | 0.097 | −0.356*** | −0.107 | 0.252** |
| **Ecosystem type** | | | | | | | |
| Alpine | Altitude | 0.104 | −0.238 | −0.224 | 0.258 | 0.232 | −0.026 |
| | Longitude | −0.107 | −0.173 | 0.071 | 0.146 | −0.093 | −0.287* |
| | Latitude | −0.164 | 0.319* | 0.181 | −0.351** | −0.207 | 0.190 |
| Subalpine | Altitude | −0.083 | 0.485* | 0.326 | −0.483* | −0.319 | 0.004 |
| | Longitude | −0.532* | 0.580* | 0.342 | −0.612** | −0.369 | 0.128 |
| | Latitude | −0.052 | 0.192 | 0.099 | −0.191 | −0.099 | 0.035 |
| Montane | Altitude | −0.306 | −0.107 | −0.264 | 0.035 | 0.223 | 0.233 |
| | Longitude | −0.221 | −0.239 | −0.309 | 0.188 | 0.282 | 0.198 |
| | Latitude | −0.137 | −0.053 | −0.338 | 0.020 | 0.320 | 0.351 |
| Valley | Altitude | 0.216 | 0.429 | 0.322 | −0.404 | −0.269 | 0.173 |
| | Longitude | 0.165 | 0.258 | 0.408 | −0.234 | −0.364 | −0.109 |
| | Latitude | 0.047 | −0.128 | −0.714** | 0.137 | 0.697** | 0.580* |
| **Functional type** | | | | | | | |
| Evergreen broadleaf | Altitude | 0.212 | 0.012 | −0.137 | 0.038 | 0.165 | 0.169 |
| | Longitude | −0.175 | −0.144 | −0.214 | 0.108 | 0.175 | 0.118 |
| | Latitude | 0.103 | 0.315* | 0.228 | −0.302* | −0.200 | 0.040 |
| Evergreen conifer | Altitude | −0.870 | 0.440 | 0.641 | −0.512 | −0.689 | −0.970* |
| | Longitude | −0.917 | 0.455 | 0.540 | −0.542 | −0.613 | −0.853 |
| | Latitude | −0.379 | −0.115 | 0.031 | 0.102 | −0.048 | −0.843 |
| Deciduous broadleaf | Altitude | −0.016 | 0.270* | 0.491*** | −0.279* | −0.498*** | −0.369** |
| | Longitude | −0.047 | 0.049 | 0.138 | −0.059 | −0.146 | −0.114 |
| | Latitude | 0.148 | −0.078 | −0.236 | 0.110 | 0.260* | 0.204 |

*, **, and *** denote significance at the 0.05, 0.01, 0.001 test level, respectively. For partial
correlation coefficients, "+" indicates a positive correlation and "−" indicates a negative correlation.
Leaf element concentrations and ratios were $\log_{10}$-transformed before analysis.





**Table A4.** Model summary for the stepwise multiple regression (SMR) of leaf element
concentrations of different ecosystem types on the climatic and soil variables (MAT,
MAP, RDI, soil element and ratio).

| Leaf element | Adj. $R^2$ Full mode | Partial regression coefficient | | | | Contribution of predictor (%) | | | |
|---|---|---|---|---|---|---|---|---|---|
| | | MAT | MAP | RDI | Soil | MAT | MAP | RDI | Soil |
| **Alpine** | | | | | | | | | |
| C | – | – | – | – | – | – | – | – | – |
| N | – | – | – | – | – | – | – | – | – |
| P | 0.059 | – | 0.001* | – | – | – | 100 | – | – |
| C:N | 0.074 | – | – | −0.037* | – | – | – | 100 | – |
| C:P | 0.066 | – | – | −0.036* | – | – | – | 100 | – |
| N:P | 0.107 | – | – | – | 0.016** | – | – | – | 100 |
| **Subalpine** | | | | | | | | | |
| C | 0.217 | – | −0.00004* | – | – | – | 100 | – | – |
| N | 0.190 | −0.010* | – | – | – | 100 | – | – | – |
| P | 0.212 | – | – | – | 0.132* | – | – | – | 100 |
| C:N | 0.201 | 0.010* | – | – | – | 100 | – | – | – |
| C:P | 0.212 | – | 0.001* | – | – | – | 100 | – | – |
| N:P | 0.554 | – | – | – | 0.036*** | – | – | – | 100 |
| **Montane** | | | | | | | | | |
| C | – | – | – | – | – | – | – | – | – |
| N | – | – | – | – | – | – | – | – | – |
| P | – | – | – | – | – | – | – | – | – |
| C:N | – | – | – | – | – | – | – | – | – |
| C:P | – | – | – | – | – | – | – | – | – |
| N:P | – | – | – | – | – | – | – | – | – |
| **Valley** | | | | | | | | | |
| C | – | – | – | – | – | – | – | – | – |
| N | – | – | – | – | – | – | – | – | – |
| P | 0.249 | 0.016* | – | – | – | 100 | – | – | – |
| C:N | – | – | – | – | – | – | – | – | – |
| C:P | 0.425 | −0.020** | – | – | 0.003* | 60.6 | – | – | 39.4 |
| N:P | 0.256 | −0.019* | – | – | – | 100 | – | – | – |

*, **, and *** denote significance at the 0.05, 0.01, 0.001 test level, respectively. For partial
regression coefficients, "+" indicates a positive correlation and "−" indicates a negative correlation.
Soil represents corresponding soil element or ratio relative to leaf element or ratio. Leaf element
concentrations and ratios were log$_{10}$-transformed before analysis.



**Table A5.** Model summary for the stepwise multiple regression (SMR) of leaf element
concentrations of different functional types on the climatic and soil variables (MAT,
MAP, RDI, soil element and ratio).

| Leaf element | Adj. $R^2$ Full mode | Partial regression coefficient | | | | Contribution of predictor (%) | | | |
|---|---|---|---|---|---|---|---|---|---|
| | | MAT | MAP | RDI | Soil | MAT | MAP | RDI | Soil |
| **Evergreen broadleaf** | | | | | | | | | |
| C | 0.231 | 0.002*** | – | – | – | 100 | – | – | – |
| N | 0.176 | – | – | – | 0.022** | – | – | – | 100 |
| P | 0.188 | – | – | – | 0.144** | – | – | – | 100 |
| C:N | 0.097 | – | – | – | 0.017* | – | – | – | 100 |
| C:P | 0.094 | – | – | – | 0.002* | – | – | – | 100 |
| N:P | 0.259 | – | – | – | 0.030*** | – | – | – | 100 |
| **Evergreen conifer** | | | | | | | | | |
| C | – | – | – | – | – | – | – | – | – |
| N | – | – | – | – | – | – | – | – | – |
| P | – | – | – | – | – | – | – | – | – |
| C:N | – | – | – | – | – | – | – | – | – |
| C:P | – | – | – | – | – | – | – | – | – |
| N:P | 0.853 | 0.012** | – | – | – | 100 | – | – | – |
| **Deciduous broadleaf** | | | | | | | | | |
| C | – | – | – | – | – | – | – | – | – |
| N | 0.075 | – | – | 0.031* | – | – | – | 100 | – |
| P | 0.251 | – | – | 0.043* | 0.159** | – | – | 43.9 | 56.1 |
| C:N | 0.069 | – | – | –0.029* | – | – | – | 100 | – |
| C:P | 0.144 | – | – | –0.061* | – | – | – | 100 | – |
| N:P | 0.093 | 0.007** | – | – | – | 100 | – | – | – |

*, **, and *** denote significance at the 0.05, 0.01, 0.001 test level, respectively. For partial
regression coefficients, "+" indicates a positive correlation and "−" indicates a negative correlation.
MAT, mean annual temperature; MAP, mean annual precipitation; RDI, standardized form of
Reconnaissance Drought Index. Soil represents corresponding soil element or ratio relative to leaf
element or ratio. Leaf element concentrations and ratios were $\log_{10}$-transformed before analysis.





**Table A6.** Model summary for the stepwise multiple regression (SMR) of leaf element
concentrations of dominant shrub species on the climatic and soil variables (MAT, MAP,
RDI, soil element and ratio).

| Leaf element | Adj. $R^2$ Full mode | Partial regression coefficient | | | | Contribution of predictor (%) | | | |
|---|---|---|---|---|---|---|---|---|---|
| | | MAT | MAP | RDI | Soil | MAT | MAP | RDI | Soil |
| ***Rhododendron telmateium*** | | | | | | | | | |
| C | – | – | – | – | – | – | – | – | – |
| N | – | – | – | – | – | – | – | – | – |
| P | – | – | – | – | – | – | – | – | – |
| C:N | – | – | – | – | – | – | – | – | – |
| C:P | – | – | – | – | – | – | – | – | – |
| N:P | 0.162 | – | – | – | 0.013* | – | – | – | 100 |
| ***Quercus monimotricha*** | | | | | | | | | |
| C | 0.732 | – | – | 0.021* | – | – | – | 100 | – |
| N | 0.700 | – | – | −0.318* | – | – | – | 100 | – |
| P | 0.904 | −0.037** | – | – | – | 100 | – | – | – |
| C:N | 0.924 | – | – | – | 0.051** | – | – | – | 100 |
| C:P | 0.919 | 0.039** | – | – | – | 100 | – | – | – |
| N:P | 0.983 | – | 0.001* | – | 0.070** | – | 32.1 | – | 67.9 |
| ***Coriaria sinica*** | | | | | | | | | |
| C | 0.885 | – | – | 0.015** | – | – | – | 100 | – |
| N | – | – | – | – | – | – | – | – | – |
| P | – | – | – | – | – | – | – | – | – |
| C:N | – | – | – | – | – | – | – | – | – |
| C:P | 0.611 | – | <0.001* | – | – | – | 100 | – | – |
| N:P | – | – | – | – | – | – | – | – | – |
| ***Bauhinia brachycarpa*** | | | | | | | | | |
| C | 0.991 | – | 0.001* | – | −0.215* | – | 7.0 | – | 93.0 |
| N | 0.997 | – | 0.018* | – | 0.169* | – | 95.7 | – | 4.3 |
| P | – | – | – | – | – | – | – | – | – |
| C:N | 0.991 | −0.373* | −0.026* | – | – | 30.1 | 69.9 | – | – |
| C:P | 0.990 | – | 0.135* | – | −7.257* | – | 17.9 | – | 82.1 |
| N:P | – | – | – | – | – | – | – | – | – |

*, **, and *** denote significance at the 0.05, 0.01, 0.001 test level, respectively. For partial
regression coefficients, "+" indicates a positive correlation and "−" indicates a negative correlation.
MAT, mean annual temperature; MAP, mean annual precipitation; RDI, standardized form of
Reconnaissance Drought Index. Soil represents corresponding soil element or ratio relative to leaf
element or ratio. Leaf element concentrations and ratios were $\log_{10}$-transformed before analysis.





**Appendix B: Additional figures**

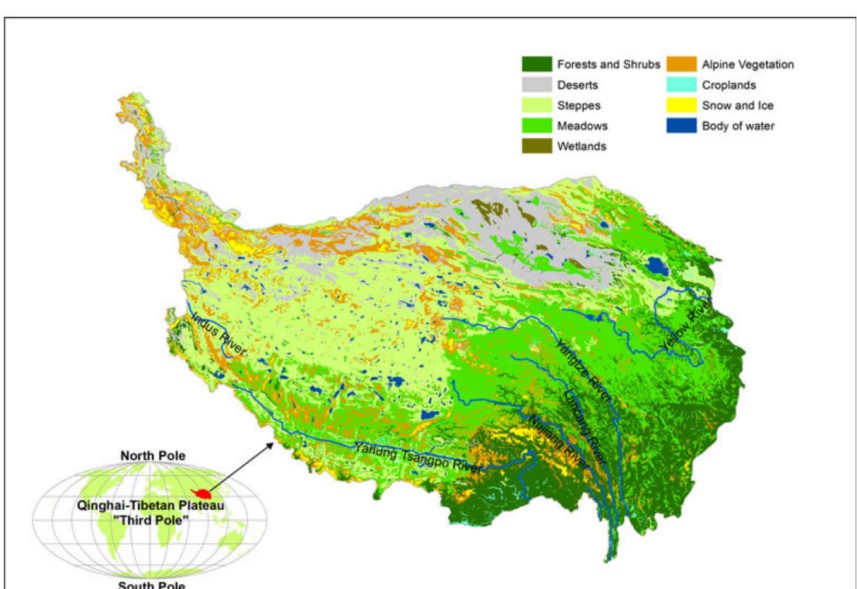


**Figure B1.** Distribution of vegetation types on the Tibetan Plateau, which is regarded
as the "third pole" of the Earth. The complex climate conditions cause a diverse
vegetation pattern, resulting in the local plant communities highly sensitive to global
climate change. The southeastern margin of the plateau is dominated by shrubs and
forests. This figure was originated from reference (Chen et al., 2013).












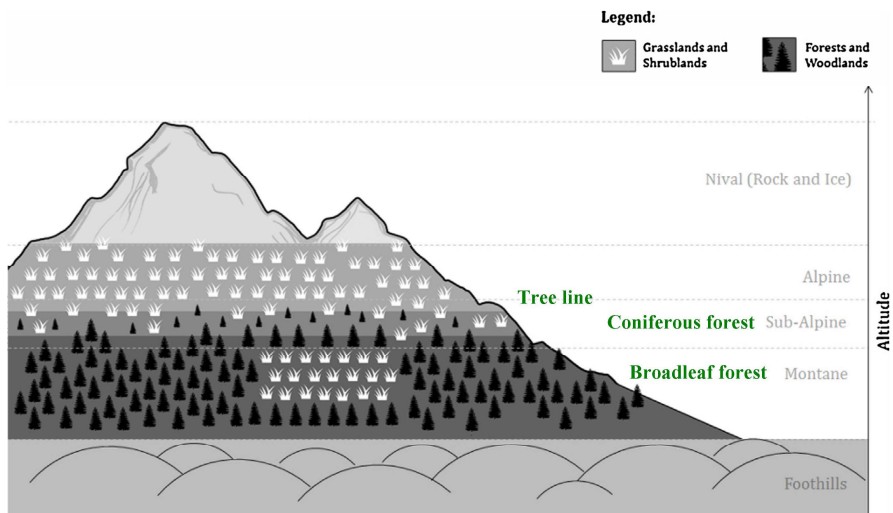

**Figure B2.** Diagram illustrating the delineation of montane, subalpine, alpine, and nival

altitudinal belts relative to the location of shrubland and grassland ecosystems. Various

shrub species are able to live in these altitudinal belts at the life form limit for shrubs,

and could not be found in the nival belt. This figure was obtained from references

(Mcavaney et al., 2001; Worboys and Good, 2011).




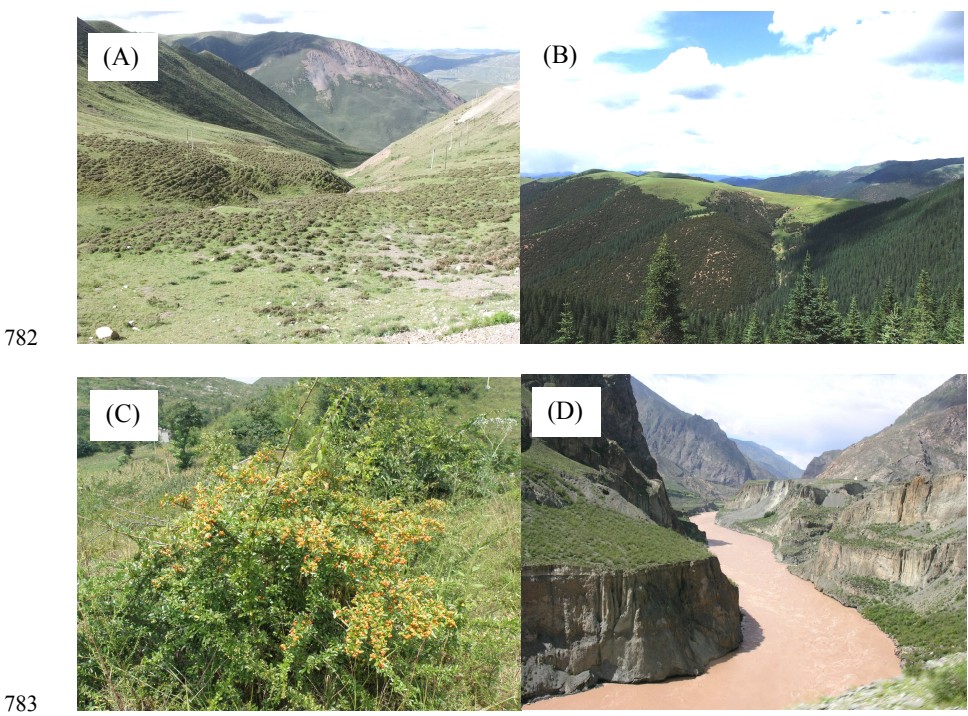


**Figure B3.** Representative photos of (A) alpine, (B) subalpine, (C) montane, and (D)
valley shrubs on the Tibetan Plateau.












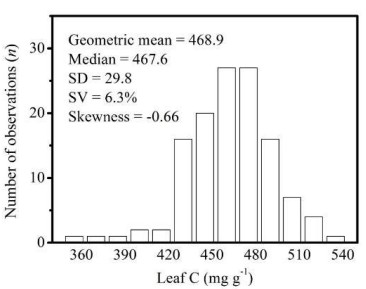


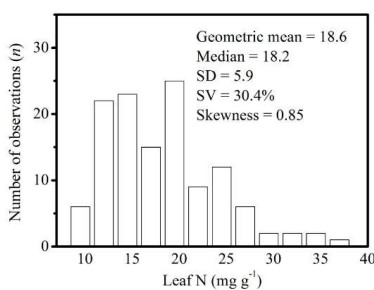

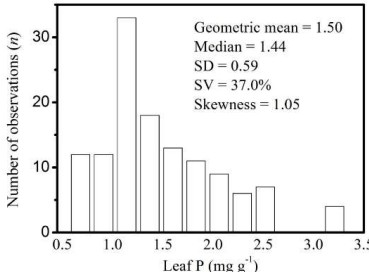


**Figure B4.** Distribution of leaf C, N and P concentrations of all shrubs on the Tibetan
Plateau.















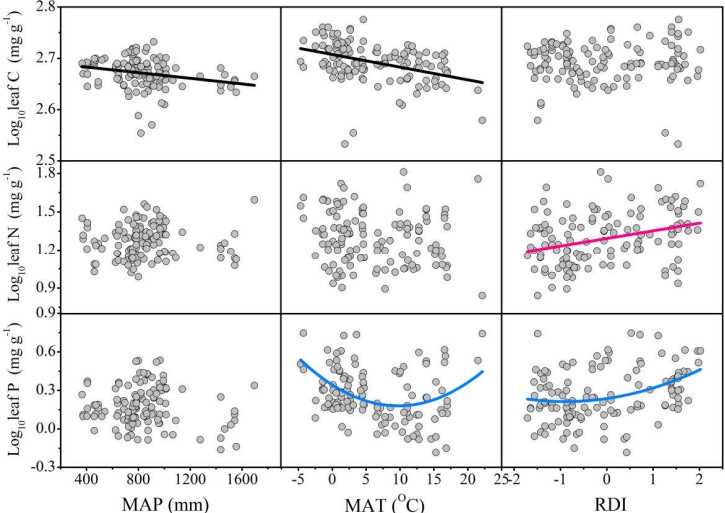


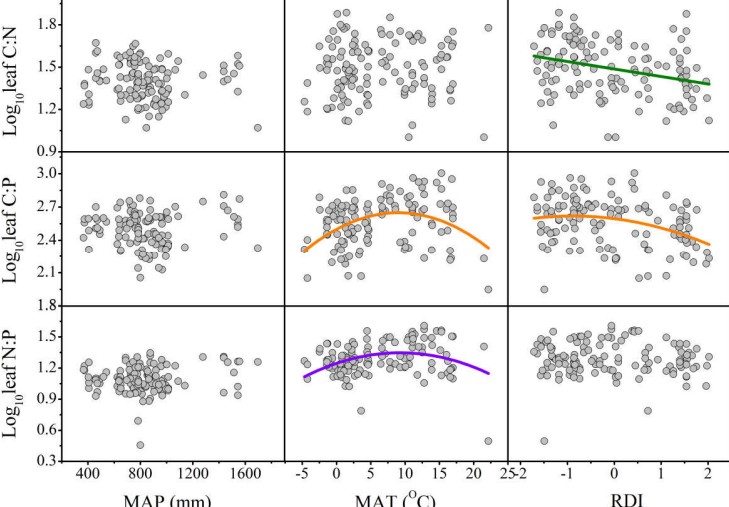


**Figure B5.** Variation trends of leaf C:N:P stoichiometry as a function of climatic factors.
Data points indicate every observation of shrub stoichiometry within the sampling sites
($n = 125$). Lines are plotted if regressions were significant at $P < 0.05$. Leaf element
concentrations and ratios were $\log_{10}$-transformed before analysis.



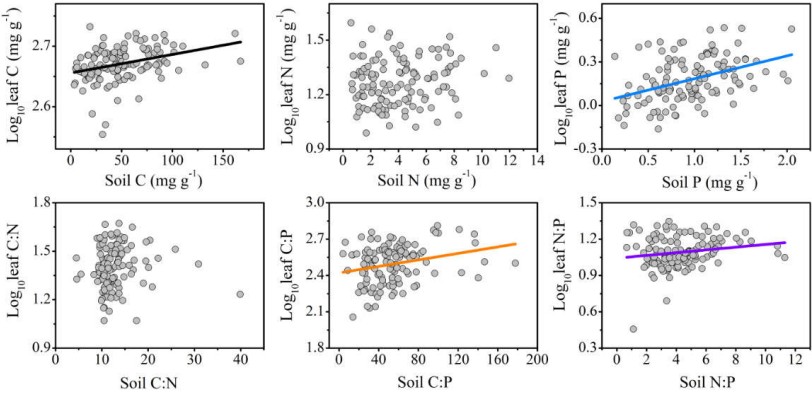


**Figure B6.** Variation trends of leaf C:N:P stoichiometry as a function of soil C:N:P.
Data points indicate every observation of shrub stoichiometry within the sampling sites
($n$ = 125). Lines are plotted if regressions were significant at $P < 0.05$. Leaf element
concentrations and ratios were $\log_{10}$-transformed before analysis.
















**Appendix C: Additional methods**
**Calculation of the Reconnaissance Drought Index (RDI$_{st}$)**
Drought severity can be evaluated via the computation of the RDI$_{st}$. RDI was proposed
by Tsakiris and Vangelis (2005), utilizing the ratios of precipitation over potential
evapotranspiration (PET) for different time scales, to be representative of the region of
interest (Khalili et al., 2011). The initial value of RDI (RDI$_\alpha$) is usually calculated for
the $i$-th year in a time basis of 12 consecutive months as follows:
$$\text{RDI}_\alpha^{(i)} = \frac{\sum_{j=1}^{12} P_{ij}}{\sum_{j=1}^{12} \text{PET}_{ij}}, \; i = 1(1)N \text{ and } j = 1(1)12 \tag{1}$$

where $P_{ij}$ and $\text{PET}_{ij}$ are the precipitation and potential evapotranspiration of the $j$-th
month of the $i$-th year, respectively, and $N$ is the total number of years of the available
data ($N = 25$ in this study). PET was calculated using the Thornthwaite method
(Thornthwaite, 1948).
As the next step, RDI$_{st}$ for a hydrological year (12-month reference period) is
computed based on the following equation:
$$\text{RDI}_{st}^{(i)} = \frac{\gamma^{(i)} - \overline{\gamma}}{\sigma_\gamma} \tag{2}$$

where $\gamma^{(i)}$ is the $\ln(\text{RDI}_\alpha^{(i)})$, $\overline{\gamma}$ is the arithmetic mean and $\sigma_\gamma$ is the standard
deviation of $\ln(\text{RDI}_\alpha)$. The RDI$_\alpha$ values are assumed to follow the lognormal
distribution, which has been found to be the most appropriate (Tsakiris et al., 2007;
Vangelis et al., 2013). The calculation process was conducted by using DrinC software.