# Peer review of "Shrub type dominates the vertical distribution of leaf C:N:P stoichiometry across an extensive altitudinal gradient"

_Biogeosciences, 2017_

## Referee Comment (RC1) · E. Bui (Referee) · 10 Jan 2018

Earlier work by Zhao et al. (2014) concluded that plant growth form, climate and soil regulate leaf C–N–P stoichiometry along an altitudinal gradient on Changbai Mountain, in NE China. This new work focuses on leaf C–N–P of shrubs in the Tibetan plateau and whether the leaf C, N, P, and stoichiometric ratios of different shrub types responds to climate and soil so it adds to the data on leaf (and soil) nutrients in China and to the body of knowledge of environmental stoichiometric patterns.

In the Discussion, landform should be acknowledged as an explanatory factor; montane and valley ecosystems occupy overlapping elevations but different landforms. Also

I think a key missing factor is the nutrient acquisition process of the shrubs–this may be why shrub type accounts for more of leaf nutrient variation than climate or soil. Is there a mycorrhizal association that helps Rhododendron telmatium acquire nutrients? Are there rhizobial nodules on Bauhinia brachycarpa?

l. 441, N-limited instead of N-limiting?

---

## Referee Comment (RC2) · Dra. Grandis (Referee) · 25 Jan 2018

Review 2: Article : Shrub type dominates the vertical distribution 1 of leaf C:N:P stoichiometry across an extensive altitudinal gradient

This work has interesting data about 125 mountainous sites that were harvested shrub leaves and was made C, N and P analysis to understanding the pattern of extensive altitudinal effects in Tibetan Plateau. The authors showed a good sampling and analysis of data that contribute with the literature about the leaf stoichiometry and shrub distribution among the mountain sites.

[Figure]

Line 65: Please correct – critical Line 238: What is CV? Coefficient variation? Please describe if is the first time appear. Line 160 and 253, 254: MAP and MAT. Please describe for the people that is not familiarized with these terms.

As a reviewer, I believe that this work has already been reviewed previously and this is a second submission. In this context, I see that it can be published. The data is interesting and shows a good presentation and a good statistical analysis. Therefore, I consider the manuscript approved.

---

## Author Comment (AC1) · 26 Jan 2018

We highly appreciate referee #1 (Dr. Elisabeth Bui) for the helpful suggestions on our manuscript. Our responses are listed below along with the corresponding changes in the text.

Response to Comments from referee #1:

Comment 1: In the Discussion, landform should be acknowledged as an explanatory factor; montane and valley ecosystems occupy overlapping elevations but different landforms. Response 1: Thank you for pointing out the effect of different landforms

between the montane and valley shrubs (Fig. R1). The explanation related to landform has been provided on lines 412−429: "It should be also noted that the valley shrub possessed the greatest leaf N and high leaf P, especially at altitude of about 1500 m. This result could be explained by the remarkably different landform of valley region from those of montane and subalpine areas with overlapping elevations, which caused distinctive local climate and shrub species in valley. On the one hand, the uplift and geological evolution of the Tibetan Plateau induced steep canyons and longitudinal range-gorge regions, with towering mountains and deep valleys arranged vertically side by side (Royden et al., 2008; Pan et al., 2012). The downvalley wind could result in foehn effect that is characterized by an increase in evaporation rate and a decrease in relative humidity (Hornsteiner, 2005; Li et al., 2007). Consequently, the MAT values of valley sites were higher than montane, subalpine and alpine sites (P < 0.05, Appendix A: Table A1), indicating faster organic matter decomposition as predicted by the Biogeochemical Hypothesis (Aerts and Chapin, 1999). On the other hand, due to the special local topography and climate, the shrubs in valley were mostly drought-tolerant plants (e.g., Bauhinia brachycarpa and Cotinus coggygria), which belonged to short-lived, fast-growing deciduous broadleaf functional type. These valley species exhibited faster litter decomposition and nutrient resorption abilities than long-lived, slow-growing evergreen types (Güsewell and Koerselman, 2002; Diehl et al., 2003)." Fig. R1 Different landforms of montane (a, b) and valley (c, d) ecosystems on the Tibetan Plateau, southwest China. Photo credit: Jun Hu. ïijĹThis figure has been provided in the Supplement.zip-Reply to referee #1.pdf) References Aerts, R., and Chapin III, F. S.: The mineral nutrition of wild plants revisited: A re-evaluation of processes and patterns, Adv. Ecol. Res., 30, 1−67, 1999. Diehl, P., Mazzarino, M. J., Funes, F., Fontenla, S., Gobbi, M., and Ferrari, J.: Nutrient conservation strategies in native Andean-Patagonian forests, J. Veg. Sci., 14, 63−70, 2003. Güsewell, S., and Koerselman, W.: Variation in nitrogen and phosphorus concentrations of wetland plants, Perspect. Plant Ecol. Evol. Syst., 5, 37−61, 2002. Hornsteiner, M.: Local foehn effects in the upper Isar Valley, part 1: Observations, Meteorol. Atmos. Phys.,

88, 175−192, 2005. Li, Y., Liu, X., Zheng, S., Chen, H., Yue, Y., Mu, C., and Liu, J.: Drought-resistant physiological characteristics of four shrub species in arid valley of Minjiang River, China, Acta Ecologica Sinica, 27, 870–877, 2007. Pan, T., Wu, S., He, D., Dai, E., and Liu, Y.: Ecological effects of longitudinal range-gorge land surface pattern and its regional differentiation, Acta Geographica Sinica, 67, 13–26, 2012. Royden, L. H., Burchfiel, B. C., and van der Hilst, R. D.: The geological evolution of the Tibetan Plateau, Science, 321, 1054−1058, 2008.

Comment 2: I think a key missing factor is the nutrient acquisition process of the shrubs–this may be why shrub type accounts for more of leaf nutrient variation than climate or soil. Is there a mycorrhizal association that helps Rhododendron telmatium acquire nutrients? Are there rhizobial nodules on Bauhinia brachycarpa? Response 2: Many thanks for the excellent suggestion on the key missing factor – nutrient acquisition strategy. This part can largely explain why the leaf nutrient levels differed among shrub species, and provide further evidence for the dominant role of shrub type in leaf element variations. We have added some discussion on lines 442−462: "Plant nutrient acquisition strategy could also affect leaf nutrient levels of different shrub species. It is well known that most plants belonging to Ericales are able to associate with soil fungi and form ericoid mycorrhiza (Perotto et al., 2002). This happens especially in high-altitude environment where plant litter decomposes slowly, leading to acidic soils rich in recalcitrant organic matter but low in available mineral nutrients (Cairney and Burke, 1998). Previous studies have reported that ericoid mycorrhiza or arbuscular mycorrhizal fungi (AMF) were associated with diverse rhododendrons in southwestern China and central Himalayan (Chaurasia et al., 2005; Tian et al., 2011). The alpine shrub species Rhododendron telmateium in similar regions probably also formed mycorrhizal fungal structures to enhance its survival and growth under stressed environments. Rhododendron telmateium may access unavailable organic N and P via the enzymatic degradation of soil organic polymers by mycorrhizal fungi (Näsholm and Persson, 2001), resulting in higher leaf N and P contents compared to Coriaria sinica. In addition, the valley shrub Bauhinia brachycarpa in our study exhibited relatively high

leaf N and P levels (although insignificant), despite its low soil total nutrients relative to alpine and subalpine shrub species (Appendix A: Table A1). This leguminous species possibly interacts symbiotically with soil microorganisms to form fungal assemblages or nitrogen-fixing root nodules, improving its nutrient acquisition in infertile soil. It has been verified that a high level of AMF diversity occurred in the rhizosphere of another dominant valley shrub species (Bauhinia faberi) on the Tibetan Plateau (Chen et al., 2016)." References Cairney, J. W. G., and Burke, R. M.: Extracellular enzyme activities of the ericoid mycorrhizal endophyte Hymenoscyphus ericae (Read) Korf & Kernan: their likely roles in decomposition of dead plant tissue in soil, Plant Soil, 205, 181–192, 1998. Chaurasia, B., Pandey, A., and Palni, L. M. S.: Distribution, colonization and diversity of arbuscular mycorrhizal fungi associated with central Himalayan rhododendrons, Forest Ecol. Manag, 207, 315–324, 2005. Chen, Y., Qu, L. Y., Ma, K. M., Yang, X. Y.: The community composition of arbuscular mycorrhizal fungi in the rhizosphere of Bauhinia faberi var. microphylla in the dry valley of Minjiang River, Mycosystema, 35, 39–51, 2016. Näsholm, T., and Persson, J.: Plant acquisition of organic nitrogen in boreal forests, Physiol. Plant, 111, 419−426, 2001. Perotto, S., Girlanda, M., and Martino, E.: Ericoid mycorrhizal fungi: some new perspectives on old acquaintances, Plant Soil, 244, 41−53, 2002. Tian, W., Zhang, C. Q., Qiao, P., and Milne, R.: Diversity of culturable ericoid mycorrhizal fungi of Rhododendron decorum in Yunnan, China, Mycologia, 103, 703–709, 2011.

Comment 3: line 441, N-limited instead of N-limiting? Response 3: The "N-limiting" has been replaced by "N-limited" on line 472.

Please also note the supplement to this comment:
https://www.biogeosciences-discuss.net/bg-2017-484/bg-2017-484-AC1-supplement.zip

---

## Author Comment (AC2) · 26 Jan 2018

We want to thank referee #2 for the helpful suggestion and positive feedback on our manuscript. Dr. Adriana Grandis is correct that this work has been previously reviewed. Our responses are listed below along with the corresponding changes in the text.

Response to Comments from referee #2:

Comment 1: Line 65: Please correct – critical. Response 1: We apologize for the mistake. This word has been revised as "critical" on line 65.

Comment 2: Line 238: What is CV? Coefficient variation? Please describe if is the

first time appear? Response 2: Yes. CV stands for the coefficient of variation. CV first appeared in part 2.5 Data analysis (line 202). We have also added this description on line 242.

Comment 3: Line 160 and 253, 254: MAP and MAT. Please describe for the people that is not familiarized with these terms? Response 3: Thanks for your suggestion. MAT and MAP have been described on lines 165–168: "In meteorology, MAT (°C) is the mean air temperature calculated by averaging the 12 months of the calendar year. MAP (mm) is the annual average value of the product of atmospheric water vapor that falls under gravity".

Please also note the supplement to this comment:
https://www.biogeosciences-discuss.net/bg-2017-484/bg-2017-484-AC2-supplement.zip